# Evaluation of Volatile Compounds in Milks Fermented Using Traditional Starter Cultures and Probiotics Based on Odor Activity Value and Chemometric Techniques

**DOI:** 10.3390/molecules25051129

**Published:** 2020-03-03

**Authors:** Li Zhang, Si Mi, Ruobing Liu, Yaxin Sang, Xianghong Wang

**Affiliations:** Department of Food Science and Technology, Hebei Agricultural University, Baoding 071001, China; zhangli3219@tom.com (L.Z.); misi@hebau.edu.cn (S.M.); koori0520@163.com (R.L.); sangyaxin@sina.com (Y.S.)

**Keywords:** volatile compounds, milk, starter cultures, characterization, heatmap analysis, principal component analysis

## Abstract

The volatile components of milks fermented using traditional starter cultures (Streptococcus thermophilus and Lactobacillus bulgaricus) and probiotics (Lactobacillus lactis, Lactobacillus bifidus, Lactobacillus casei, and Lactobacillus plantarum) were investigated by means of gas chromatography-mass spectrometry (GC-MS) combined with simultaneous distillation extraction (SDE). A total of 53 volatile compounds were detected, being 10 aldehydes, 11 ketones, 10 acids, 11 hydrocarbons, 7 benzene derivatives, and 4 other compounds. The starter culture was found to significantly affect the composition of volatile components in the fermented milks. Ketones and hydrocarbons were the dominant compounds in milk before fermentation, while acids were dominant compounds in the fermented samples. Compared with probiotics, there was greater abundance of volatile components in fermented milks with traditional strains. The importance of each volatile compound was assessed on the basis of odor, thresholds, and odor activity values (OAVs). Of the volatile compounds, 31 of them were found to be odor-active compounds (OAV > 1). The component with the highest OAVs in most samples was (E,E)-2,4-decadienal. Heatmap analysis and principal component analysis were employed to characterize the volatile profiles of milks fermented by different starter cultures. The results could help to better understand the influence of starter cultures on the odor quality of milks.

## 1. Introduction

Fermentation is a common method of producing new and palatable dairy products from milk [1]. Fermented milk is widely consumed not only due to rich nutrition but also for its sensory properties. The primary sensory attributes of fermented milk include texture, color, taste, and odor [2]. Among these attributes, odor has played an important role in determining the acceptability and preference of fermented milk for customers. It is caused by one or more volatile compounds, generally, at a very low concentration level, that humans or other animals perceive by the sense of olfaction [3]. As an important fermented milk, yogurt has been extensively studied for its volatile components. Studies suggest that there are more than 90 different volatile compounds in yogurt, including carbohydrates, alcohols, aldehydes, ketones, acids, esters, lactones, sulphur-containing compounds, pyrazines, and furan derivatives [4,5,6].

The flavor of fermented milk is formed by the action of starter bacteria and originated from biochemical changes of carbohydrates, lipids, and proteins [7]. Starter cultures produces volatile substances that contribute to the typical flavor of certain fermented products, such as yogurt (acetaldehyde), butter, and buttermilk (diacetyl), as well as kefir and koumiss (ethanol) [1]. Currently, two species, including *Streptococcus thermophilus* and *Lactobacillus bulgaricus,* are widely used for the industrial fermentation of yogurt. Many studies have been performed to investigate the factors affecting the volatile composition of yogurt, such as extraction techniques [8], source of milk [9,10], packaging materials [11], processing [12], and storage conditions [10,13].

In recent years, along with the increase of consumers’ demand for nutritious and healthy yogurts, more and more probiotics have been used in milks fermentation. It has been well-reported that probiotics are associated with such health benefits as prevention/treatment of hypercholesterolemia [14], certain cancers [15,16,17], immune diseases [17,18,19], anti-obesity [20,21], and insulin resistance [22,23]. When probiotics are added to fermented milk, they can play an important role in the product odor quality. Zareba et al. (2012) determined that *Bifidobacterium* fermentation increased low-molecular volatile compounds (acetic acid, 2-pentanone, and 2-butanone) in milk [24]. Pan et al. (2013) analyzed the volatile compounds in milk fermented by *Lactobacillus pentosus* and confirmed that butanoic acid, methyl ester, octanoic acid, and decanoic acid were the main volatile compounds in the sample [25]. However, there are few studies focusing on the difference of in odor quality of milks fermented by different starter cultures, especially difference between traditional starter cultures and probiotics.

In addition, not all difference of the volatile components found in fermented milks are of sensory importance, as these components differ in concentration. Only when a volatile compound concentration exceeds an odor threshold can the components can be perceived [26]. Measuring the odor active value (OAV) is an effective method to evaluate odor intensity by using the ratio of the concentration of a certain odor to its threshold value. When a volatile compound’s OAV is greater than 1, it contributes to the overall odor of a sample [27]. Therefore, OAV was widely introduced to choose impact odorants in dairy products, such as milk [28,29], milk powder [30,31], and cheese [32,33]. Tian et al. (2019) reported that two *Lactobacillus lantarum* strains (1-33 and 1-34) could improve the odors of acetaldehyde, diacetyl, and acetoin based on OAVs in yogurt [34]. However, there are few studies concerning the difference between traditional starter cultures and probiotics fermentation on the milks’ odor quality.

Chemometric techniques are of extreme importance and widely relevant in the resolution of problems in flavor chemistry. Among them, hierarchical cluster analysis (HCA) and principal component analysis (PCA) have often been applied in interpreting the chromatographic data related to an extensive number of compounds and a large number of variables involved. PCA and HCA are exploratory multivariate data analysis techniques. PCA is a projection technique, the goal of which is to reduce large amounts of data and represent them efficiently in a graphical form. HCA is a clustering technique complementary to PCA, which groups a dataset by similarities or dissimilarities [35].

The objective of this study was to characterize the volatile profiles of milks fermented by different starter cultures including *Streptococcus thermophilus* s2 (STs2), *Lactobacillus bulgaricus* 8 (LB8), *Lactobacillus lactis* 28 (LL28), *Lactobacillus bifidus* 531 (LB531), *Lactobacillus casei* 475 (LC475), and *Lactobacillus plantarum* 45 (LP45). Gas chromatography-mass spectrometry (GC-MS) combined with simultaneous distillation extraction (SDE) was applied to analyze the composition of volatiles in samples. Chemometrics was performed by heatmap analysis and PCA to screen characteristic volatiles of different fermented milks. This study could be important for fostering a better understanding of typical volatile compounds in fermented milks to improve the quality of product.

## 2. Results and Discussion 

### 2.1. Composition of Volatiles in Milks Fermented by Different Starter Cultures

#### 2.1.1. Distribution of Volatiles in Milks Fermented by Different Starter Cultures

Relative concentration and compound amount of different type volatile compounds identified in milk before fermentation (BFM); traditional starter cultures (STs2 and LB8); and probiotics (LL28, LB531, LC475, and LP45) samples are shown in Figure 1. A total of six types of volatile compounds were detected, including aldehydes, ketones, acids, benzene derivatives, hydrocarbons, and other compounds. As shown in Figure 1A, ketones (38.41%) and hydrocarbons (49.56%) were the predominant volatile components in BFM samples, whereas acids (ranged from 41.95% to 80.48%) were the main constituents in the fermented samples. In addition, for BFM, STs2, LB8, LL28, LB531, LC475, and LP45 samples, 35, 35, 35, 28, 31, 27, and 29 volatile compounds were found, respectively (Figure 1B). High amounts of ketones (7–9) were detected in all samples. However, high amounts of aldehydes and benzene derivatives were detected in BFM samples, and high amounts of acids (6−10) were detected in fermented samples. Compared with probiotics, there was greater abundance of volatile components in fermented milks with traditional strains (STs2 and LB8). The abundant volatile components and excellent synergistic fermentation might be the important reasons why STs2 and LB8 were selected as industrial strains used in yogurt production.

#### 2.1.2. Volatiles in Milks Fermented by Different Starter Cultures

The volatile compounds identified in BFM, fermented samples are shown in Table 1. A total of 53 volatile compounds were detected, including 10 aldehydes, 11 ketones, 10 acids, 7 benzene derivatives, 11 hydrocarbons, and 4 other compounds. The total concentrations of volatiles ranged from 1042.44 to 4987.95 mg·kg^−1^, from low to high in BFM, LB8, LP45, LL28, STs2, LB531, and LC475 samples, respectively. The total concentrations of volatile compounds in STs2 and probiotics were significantly higher than BFM and LB8 samples (*p* < 0.05). Of these volatiles, 16 were commonly detected in all investigated samples, and 21 of them were found to be simultaneously present in the milks fermented by different starter cultures. 

Ten aldehyde compounds were detected in the volatile fraction of milks fermented by different starter cultures. Among the detected aldehydes, three compounds belonged to aldehydes containing benzene, six aldehydes belonged to *n*-alkanals, and one aldehyde belonged to *2,4*-alkadienals. Eight, four, eight, four, four, three, and five aldehydes were detected in BFM, STs2, LB8, LL28, LB531, LC475, and LP45 samples at different concentrations of 36.64, 9.46, 35.60, 11.00, 13.07, 12.57, and 14.52 mg·kg^−1^, respectively (Table 1). Among of all samples, aldehyde concentrations in BFM and LB8 samples were significantly higher than other samples (*p* < 0.05). Aldehyde concentrations were relatively low, but the number and content of aldehydes were significantly different. Benzaldehyde, nonanal, and (E,E)-2,4-decadienal were detected in all samples. Octanal was only detected in BFM and STs2. Benzeneacetaldehyde was detected in LB8 and LL28 samples. Decanal, 2,4-dimethyl-benzaldehyde, and dodecanal were only in BFM and LB8 samples, and undecanal was only in LB8, LB531, and LP45. Tetradecanal was detected in just BFM samples. Aldehydes can be derived from amino acid catabolism [36] or the decomposition of hydroperoxides, an oxidation product of unsaturated fatty acids [37]. In addition, aldehydes can be reduced to alcohols by alcohol dehydrogenases or oxidized to carboxylic acids by aldehyde dehydrogenase [38]. Therefore, due to related enzyme activity differences, there were changes in aldehyde content in fermented milks from different starters. It is a topic which needs further study.

A total of 11 ketones were detected in all samples. Eight, nine, eight, eight, nine, seven, and eight ketones with concentrations between 340.91 and 494.48 mg·kg^−1^ were in BFM, STs2, LB8, LL28, LB531, LC475, and LP45 samples, respectively (Table 1). Seven ketones belonged to methyl ketones, one ketone belonged to unsaturated detected ketones, and three ketones belonged to heterocyclic ketones. Six ketones, including 2-heptanone, 2-nonanone, 2-undecanone, 2-tridecanone, 2-pentadecanone, and 6-heptyltetrahydro-2H-pyran-2-one, presented in all samples. In the BFM sample, ketones were the dominant compounds (38.41%). However, in the fermented samples from different starter cultures, there was little change in total ketone content, and they were not a dominant volatile component (between 8.30% to 21.04%). Ketones are inherently important volatile ingredients in milk, and the preheating treatment during yogurt processing may accelerate ketone formation in milk [31]. Ketone compounds can be generated from autoxidation reactions of unsaturated fatty acids during fermentation [31]. In addition, changes in ketones during fermentation is affected by textural properties, stabilizing agents, and the concentration of fatty acids [39].

Ten acids were detected in all samples. In the BFM sample, only two acids (dodecanoic acid and tetradecanoic acid) with a concentration of 11.90 mg·kg^−1^ were detected. After fermentation, there was a significant increase in acid quantity and content. Ten, six, eight, nine, eight, and eight acids with concentrations of 3016.65, 937.95, 3016.97, 3542.22, 4013.10, and 3043.45 mg·kg^−1^ were detected in STs2, LB8, LL28, LB531, LC475, and LP45 samples, respectively (Table 1). After fermentation by different starter cultures, acids became the dominant volatile compounds in samples. The acid concentration was the highest in the LC475 sample. The number of acids was the highest in STs2 sample, in which all ten acids were detected. The number (six acids) and acid concentration (937.95 mg·kg^−1^) were the lowest in the LB8 sample. As shown in Table 1, the same eight acids were detected in LL28, LC475, and LP45 samples. Octanoic aid, decanoic acid, dodecanoic acid, and tetradecanoic acid, which concentrations usually exceeded 5%, were the main acids in fermented samples. The hydrolysis of triacylglycerols under the action of lactic acid bacteria is the main pathway for the formation of carboxylic acids [40]. Carboxylic acids are not only aroma components but also precursors for the formation of methyl ketones, alcohols, lactones, aldehydes, and esters [25].

Eleven hydrocarbons were detected in all samples. Seven, eight, six, four, five, five, and four hydrocarbons with concentrations between 302.05 to 631.48 mg·kg^−1^ were in BFM, STs2, LB8, LL28, LB531, LC475, and LP45 samples, respectively (Table 1). Among the 11 hydrocarbons, nonane was detected among the predominant hydrocarbons, with concentrations ranging from 297.29 to 621.57 mg·kg^−1^. The other hydrocarbons with low content also were detected in all samples. Hydrocarbons are generally the secondary products of lipid autoxidation in raw milk [41]. Compared to two industrial commonly used strains, STs2 and LB8, a low content of nonane was detected in the probiotics of fermented samples. Hydrocarbons may not contribute to samples aroma because of their high perception threshold [41]. They can serve as precursors for the formation of other volatile compounds [41]. The decrease of hydrocarbons in fermented samples may be due to their role as intermediate for other compounds under the action of starter cultures.

Seven benzene derivatives were detected in low concentrations in samples by different starter cultures. Seven, three, four, three, three, three, and three benzene derivatives were detected in BFM, STs2, LB8, LL28, LB531, LC475, and LP45 samples at different concentrations of 36.56, 38.98, 35.6, 30.59, 26.02, 22.0, and 28.49 mg·kg^−1^, respectively (Table 1). Fermentation can significantly reduce the amount of benzene derivatives in fermented milks (*p* < 0.05). Among detected benzene derivatives, 1,2,4,5- tetramethyl-benzene, naphthalene, 1-methyl-naphthalene, and 2,3-dimethyl-naphthalene could not be detected in fermented samples. Naphthalene has only been detected in LB8 fermented milk. Benzene derivatives usually come from the milk of cows fed on pasture grass [42]. The ability of fermentation to reduce the number of benzene derivatives may be related to a unique biological activity of the microorganism, leading to degradation of benzene derivatives [43].

In addition, there were four other compounds detected, including 2-pentyl-furan, limonene, benzothiazole, and 2,4-bis(1,1-dimethylethyl)-phenol in fermented milks from different starter cultures. Most of them are common volatile compounds in yogurt. The compound 2-pentyl-furan only presented in BFM samples, which is a secondary oxidation product of linoleic acids [39]. Limonene and benzothiazole were detected in BFM and LB8 and LB8 samples, respectively. These compounds may come from a cow’s diet, which is transferred to milk [42]. The compound 2,4-bis (1,1-dimethylethyl)-phenol was detected in all samples, but fermentation significantly (*p* < 0.05) reduced the concentration of 2,4-bis(1,1-dimethylethyl)-phenol. The reason for this concentration decrease is unknown and needs further study.

### 2.2. Composition of Odor-Active Compounds in Milks Fermented by Different Starter Cultures

#### 2.2.1. Distribution of Odor-Active Compounds in Milks Fermented by Different Starter Cultures 

Because different volatile compounds have different thresholds, not all components have a large impact on the overall aroma character of fermented milks from different starter cultures. The OAV is an important method to evaluate the contribution of various volatile compounds on the olfactory impression. Compounds with OAV > 1 were identified as odor-active compounds. Figure 2 shows distribution of odor-active compounds in milks fermented by different starter cultures based on OAV. Nineteen, 20, 21, 18, 18, 16, and 19 with OAVs > 1 were detected in BFM, STs2, LB8, LL28, LB531, LC475, and LP45 samples, respectively (Figure 2A). Compared with probiotics, there were greater abundance odor-active compounds detected in fermented milks with traditional strains (STs2 and LB8). According to the level of OAVs, OAVs are classified into four categories. OAVs which are more than 1000 are extremely high OAVs. OAVs which are between 100 and 1000 are high OAVs. OAVs which are between 10 and 100 are medium OAVs, and OAVs which are between 1 and 10 are low OAVs. As shown in Figure 2B, there were 3-4, 5-7, 5-8, and 2-5 compounds detected to belong to the extremely high, high, medium, and low OAVs compounds, respectively. Different strains of fermentation had obvious effects on odor intensity in fermented milks.

#### 2.2.2. Odor-Active Compounds in Milks Fermented by Different Starter Cultures 

The aroma characteristics of 32 volatile compounds were described in previous works [26,34,44]. As shown in Table 2, OAVs were >1 in 31 volatile compounds. These compounds played a critical role in milks fermented by different starter cultures. Among odor active characteristic components, eight components, including (E,E)-2,4-decadienal, 2-heptanone, 2-nonanone, 2-undecanone, dodecanoic acid, ethylbenzene, p-xylene, and styrene, were detected in all samples.

As shown in Table 2, aldehydes made up the most important group of odor-active volatiles due to lower thresholds and high OAVs. Among the 10 detected aldehydes, eight aldehydes belonged to odor active compounds. Most of them presented high OAVs and have an important effect on the odor quality of the milks fermented by different starter cultures. Octanal, (E,E)-2,4-decadienal, and dodecanal presented extremely high OAVs and greatly influenced the flavor of samples. (E,E)-2,4-decadienal should be noted. Its OAV exceeded 10,000 at 20,301 (LL28) and 43,761 (LB8), and it was the most important odor active compound in samples. Due to an extremely low odor threshold of (E,E)-2,4-decadienal (0.0001 mg·kg^−1^, “fatty and green”), trace changes had a huge impact on odor intensity of samples. Octanal contributed fat, soap, lemon, and green odor and presented extremely high OAVs in BFM (1608), STs2 (1035), and LP45 (1302). Dodecanal contributed “soapy, waxy, and citrus” odor, and presented 5836 and 7324 OAVs in BEM and LB8 samples, respectively. Benzene acetaldehyde (LB8 and LL28); decanal (BFM, LB8, LL28, LC475, and LP45); and undecanal (LB8 and LB531) showed high OAVs and contributed “sweet and floral”, “fatty”, and “fatty” odors to samples, respectively. Nonanal (sweet, floral, citrus, and grass) and decanal (fatty) were identified as medium OAVs in STs2, LB531, and BFM LB8 samples. Benzaldehyde belonged to low OAVs and was present in almost all samples except LC475 samples. Among detected odor-active aldehydes, nonanal and decanal were detected in milk fermented by *Lactobacillus plantarum*, but they did not contribute odor to a sample’s OAV [34]. Nonanal, (E)-decanal, and (E,E)-2,4-decadienal were reported to contribute “fatty”, “fruity”, and “fruity” odors respectively in fermented camel milk [8]. In addition, octanal and decanal contribute “fatty, green” and “green” odors to dairy product such as cheese [45].

Six ketones were found to affect aroma characteristics of samples from different starter cultures. Odor-active ketones belonged to second-odor-strength compounds, which were inferior to aldehydes. The ketone 2-none-4-one, which contributed “fruity and floral” odors, showed extremely high OAVs in STs2 (3391), LB8 (33047), LL28 (1915), LB531 (3939), and LP45 (1917) samples due to their low threshold (0.0009 mg·kg^−1^). “Fruity and floral” are pleasant odors for people. The LB8 sample had extremely high OAVs (33047), which are about 10 times higher than other starters. This may be why it is often selected as an industrial strain and used together with *Streptococcus thermophilus*. The ketone 2-heptanone contributed a “fresh cream” odor with high OAVs between 386 and 815. The ketones 2-nonanone and 2-undecanone, which had extremely high OAVs or high OAVs in some samples, contributed “grassy, fruity, and floral” and “orange, grass, and fresh” to samples. Ketones 2-decanone (sweet and waxy) and 2-dedecanone (orange, grass, and fresh) with medium OAVs only contributed odor in the BFM sample. Among the odor-active ketones, 2-heptone, 2-nonanone, and 2-undecanone contributed medium OAVs of “fresh cream”, “hot milk and foam”, and “foam and grass” to yogurt fermented by *Lactobacillus plantarum* [34]. The ketone 2-decanone, which contributed “fresh cream”, was identified as an odor active compound in fermented camel milk [8].

Among acids with an odor description, most of them had OAVs >1, except for tridecanoic acid, and mainly contributed odor in fermented samples. In the BFM sample, only dodecanoic acid provided a “rancid” odor with low OAV (1.1), which had a slight contribution to sample odor. Octanoic acid, decanoic acid, dodecanoic acid, and tetradecanoic acid, which provide “chess”, “sweaty”, and “rancid” odors, were detected as odor-active compounds and were classified as high OAVs or medium OAVs. However, compared to other fermented samples, milk, which was fermented by LB8, showed lower OAVs. Hexanoic acid (spicy, rancidity, and floral); nonanoic acid (rancid); and undecanoic acid (rancid) with medium OAVs were detected in fermented samples expected for LB8 samples. Heptanoic acid and pentadecanoic acid with low OAVs only were detected in STs2-fermented milk. Acid compounds are important characteristic odor components in fermented milks. These detected acids also are odor active components in dairy products. Tian et al. (2019) reported that hexanoic acid and octanoic acid with “spicy, rancidity, and floral” and “chess” characteristic odors were detected as odor-active compounds in yogurt which were fermented by *Lactobacillus plantarum* with medium OAVs [34]. In addition, undecanoic acid, decanoic acid, dodecanoic acid, and tetradecanoic acid provided odors such as “fatty, sweet, floral, and waxy” in dairy products such as fermented camel milk [8] and cheese [45].

Five odor-active benzene derivatives, including ethylbenzene, p-xylene, styrene, naphthalene, and 1-methyl-naphthalene, gave a “phenolic” odor to samples. OAVs of ethylbenzene, *p*-xylene, and styrene showed medium odor intensity and high odor intensity in all samples, respectively. Naphthalene and 1-methyl-naphthalene presented high odor intensity in BFM samples. Among all fermented samples, only naphthalene was detected in LB8 samples as an odor-active compound with high OAVs. Benzene derivatives such as ethylbenzene and *p*-xylene were reported as providing “phenol, spice” and “leathery” odors to yogurt [34]. Other compounds were detected as active odor compounds in samples. Among them, 2-pentyl-furan provided a “green, fat” odor to BFM samples with high OAVs. Limonene showed medium OAVs and low OAVs in BFM and LB8 samples, respectively. Benzothiazole with low OAVs only was detected in the LS8 sample.

### 2.3. Heatmap Analysis of Volatiles Profiles of Milks Fermented by Different Starter Cultures

#### 2.3.1. Heatmap Analysis of Volatile Compounds of Milks Fermented by Different Starter Cultures Based on the Content

A heatmap was generated to show the variations in the contents of volatile compounds in milks fermented from different starter cultures (Figure 3). Green colors indicate that volatile compound levels were less than their mean levels, while red colors indicate that the volatile compound levels were higher than their mean levels. The samples collected from different starter cultures fermentation were clustered into two clusters; the first cluster contained BFM and LB8 samples and the second cluster contained STs2 probiotics (LL28, LB531, LC475, and LP45) samples. In the second culture, STs2 belonged to one subcluster, and four probiotics belonged to another subcluster. Based on the volatile compound content, volatile compound s were divided into three groups: A, B, and C. The A group included five ketones, nine acids, one aldehyde, and three hydrocarbons. Almost all acids and some ketones were in this group. They showed relatively high content in STs2, LL28, LB531, LC475, and LP45 samples and represented the typical volatile compounds in fermented samples. These compounds usually result from the lipid degradation. For example, acids are product of hydrolysis of triacylglycerols under the action of starter cultures [40], and ketones are generated from auto-oxidation reactions of unsaturated fatty acids [31]. B group included eight compounds: benzeneacetaldehyde, pentadecanoic acid, ethylbenzene, *p*-xylene, 1-heptadecene (E,E)-2,4-decadienal, 2-nonen-4-one, and benzothiazole. This group showed high concentrations in STs2 and LB8, both of which are lactic acid bacteria commonly used in industry. These volatile compounds may be the typical volatile compounds in daily yogurt and may reflect the main differences between industrial strains and probiotics. C group, which showed high content in BFM samples, included six aldehydes, five ketones, seven hydrocarbons, five benzene derivatives, and three other compounds. Most hydrocarbons, benzene derivatives, and other compounds, and parts of aldehydes and ketones, were concentrated in the C group. These compounds represent volatile compounds in raw milk. For example, benzene derivatives are representative compounds that come from raw milk [42]. Hydrocarbons are generally the secondary products of lipid autoxidation in raw milk [41].

#### 2.3.2. Heatmap Analysis of Volatile Compounds of Milks Fermented by Different Starter Cultures Based on OAVs

Figure 4 shows a heatmap of variations in the OAVs of volatile compounds in milks fermented from different starter cultures. STs2 and LB8 belonged to the 1st cluster, and STs2 probiotics (LL28, LB531, LC475, and LP45) samples belonged to the second cluster. In the second cluster, probiotics (LL28, LB531, LC475, and LP45) samples belonged to one subcluster in the OAVs heatmap. According to the OAVs, which presented the odor intensity of the volatile compounds, odor-active compounds were divided into four groups: A, B, C, and D. A group contained six compounds, which showed relatively high odor intensities in the second cluster samples, including two aldehydes, two ketones, and seven acids. They showed relatively high odor intensities in the STs2 samples and probiotic (LL28, LB531, LC475, and LP45) samples. In this group, “rancid”, which came from acids is the typical odor characteristics of fermented milk. B group contained heptanoic acid, pentadecanoic acid, ethylbenzene, and *p*-xylene, which showed high odor intensity in LB8 and STs2 samples. Acids and benzene derivatives provided “rancid” and “phenolic”, which expressed odor characteristics of two industrial strains [34,44]. There were 12 odor active compounds, including four aldehydes, three ketones, three benzene derivatives, and two other compounds in the C group. These odor-active compounds mainly provided “fatty”, “fruity”, and “flora”, which came from aldehydes, ketones, and benzene derivatives and other compounds [26,34,44]. They represent the odor characteristics of unfermented milk. Most aldehydes and benzene derivatives focused this group and contributed high odor intensity in SFM samples. Benzeneacetaldehyde, 2-nonen-4-one, (E,E)-2,4-decadienal, and benzothiazole presented high odor in LB8 samples and were grouped in the D group. There were similar distributions in milk-fermented different starter cultures in the heatmap of concentrations and OAVs of volatile compounds.

### 2.4. PCA of Volatile Profiles of Milks Fermented by Different Starter Cultures

#### 2.4.1. PCA of Volatile Compounds of Milks Fermented by Different Starter Cultures Based on the Content

PCA was applied to the concentrations of 53 volatile compounds to analyze differences among milks fermented from different starter cultures (Figure 5). In Figure 5A, the score scatter plot for the two first principal components (PC1 and PC2) represents the differences among the samples. Figure 5B represents the corresponding loading plot, which established the relative importance of each volatile compound and the relationships between volatile compounds and samples. PC1 and PC2 explained approximately 58.82% and 17.34% of the total variation, respectively (76.16% collectively). The samples collected from different starter culture fermentations were clustered into three groups. The first group was laid on the positive side of PC1 and included STs2 and probiotics (LL28, LB531, LC475, and LP45) samples. The second group only contained the LB8 sample, and the third group only contained the BFM sample. This distribution of PAC was similar to the distribution of the heatmap in Figure 3, since both of them were divided into three groups. In addition, the composition of every group also was similar. STs2 and probiotics (LL28, LB531, LC475, and LP45) samples were located in the positive region of PC1 and the negative region of PC2. There were six acids, two hydrocarbons, and two ketones which were located in this region. These samples had strong correlation with acid compounds, including hexanoic, heptanoic, decanoic, undecanoic, dodecanoic, and tridecanoic acid. They are typical compounds in fermented milk and belong to the product of hydrolysis of triacylglycerols under the action of lactic acid bacteria [40]. Sample LB8, which was independent of the other fermentation stains, was located in the negative region of PC1 and the positive region of PC2. There were total of eight compounds in this region. It was strongly correlated with volatile compounds, such as nonanal, decanal, naphthalene, and limonene. The sample BFM was located in the negative region of PC1 and the negative region of PC2, and these regions contained five ketones, four aldehydes, four benzene derivatives, five hydrocarbons, and two other compounds. Most of compounds which located in this region represent the volatile compounds in unfermented samples. Ketones, benzene derivatives, and hydrocarbons are representative compounds that come from raw milk [42].

#### 2.4.2. PCA of Volatile Compounds of Milks Fermented by Different Starter Cultures Based on the OAVs

PCA also was applied to the concentrations of 31 odor activity characteristics compounds (OAV > 1) to analyze differences among milks fermented from different starter cultures (Figure 5). The PC1 and PC2 regions explained approximately 51.36% and 22.88% of the total variation, respectively (74.24% collectively). This distribution of volatile profile (Figure 6A) was consistent to the distribution of the volatile profile in Figure 5A based on OAVs. STs2 and probiotics (LL28, LB531, LC475, and LP45) samples were located in the fourth quadrant of the PCA plot. There were six acids in this region, and the “rancid” note associated with these acids [34,44] was found to be correlated with STs2 and probiotics (LL28, LB531, LC475, and LP45) samples. LB8 sample lay in the second quadrant of the PCA plot. Four aldehydes, two benzene derivatives, one ketone, one acid, and one other compound were located in this quadrant. The “fatty” note associated with decanal, the “soapy, waxy, and citrus” notes associated with dodecanal, and the “phenolic” note associated with naphthalene [34,44] were found to be correlated with LB8. (E,E)-2,4-decadienal provided a “fatty, green” odor, which was the dominant odor in all samples. BFM samples lay in the third quadrant of the PCA plot. There were two aldehydes, three ketones, two benzene derivatives, and one other compounds in this third quadrant. The “sweet”, “orange”, and “grass” notes associated with nonanal, 2-decanone, 2-dodecanone, and limonene [26,34]; the “green” and “fat” notes associated with 2-pentyl-furan [32]; and the “phenolic” note associated with 1-methyl-naphthalene [44] were found to be correlated with BFM. Similar PCA distributions were found in milks fermented by different starter cultures based on the content and OAVs of volatile compounds. *S. thermophilus* is a multifunctional lactic acid bacteria also used as a probiotic in some studies. As shown in heatmap and PCA, *S. thermophilus* behaved more like other probiotic strains. The different distribution of odor components may be caused by the probiotic’s activity. The inherent relationship between probiotic activity and volatile components needs further study.

The heatmap and PCA showed a highly consistent distribution of volatile profiles based on the content and the OAVs in this research. Samples collected from different starter culture fermentations were clustered into three groups. STs2 and LL28, LB531, LC475, and LP45 samples belonged in one kind. LB8 and BFM belonged to the second and third groups and presented a closer distribution trend. LB8 and STs2 presented a further distribution; however, they can synergistically enhance the odor quality of fermented milk. All of these reasons may explain why they have been selected as industrial fermentation strains. However, synergistic action in further research is necessary to study yogurt odor mechanisms. *S. thermophilus* is a multifunctional lactic acid bacteria also used as a probiotic in some studies [46]. As shown in heatmap and PCA, *S. thermophilus* behaved more like other probiotic strains. The different distribution of odor components may be caused by the probiotic’s activity. The inherent relationship between probiotic activity and volatile components needs further study.

## 3. Materials and Methods 

### 3.1. Fermented Milks Preparation

Pasteurized full-fat milk from a local retailer (Hebei Sanyuan Food Co., Ltd., Hebei, China) was fortified with 6% sugar. Homogenization was performed at 5 pa, divided into 500 mL glass bottles, and then heated at 98 °C for 15 min. Milk was then immediately cooled to about 35 °C in a water bath and inoculated with different starter cultures (Hebei Inatural Bio-Tech Co., Ltd., Hebei, China). Table 3 shows the fermentation conditions. When pH reached 4.5, the samples reached the end of fermentation. After fermentation, the samples were placed in a refrigerator at 4 °C for 24 h to age. Finally, milks fermented by different starter cultures were stored at −24 °C until subsequent analysis. Milk before fermentation (BFM) was also kept at −24 °C for analysis. Extraction of each sample was performed in triplicate.

### 3.2. Extraction of Volatile Compounds

The volatile compounds of samples were extracted using SDE in a Likens-Nickerson apparatus. A mixture of 200 g of sample and 2 L of deionized water was placed in a 5 L flask and connected to a Likens-Nickerson apparatus. Aqueous 2, 4, 6-trimethylpyridine (40 mg·kg^−1^) was added as an internal standard. Then, an U-tube apparatus was filled with water and 50 mL diethyl ether. An additional 10 mL diethyl ether was poured into a finger-shaped-bottom flask, and both flasks (one containing diethyl ether and another containing the sample) were heated to boiling. The extraction time was 2 h. After extraction, the distillate in the 100 mL flask was dried over anhydrous sodium sulfate (5 g), concentrated using vacuum rotary evaporation, and stored in headspace vials. 

### 3.3. GC-MS Analysis

The analysis of volatile compounds was carried out using a GC system (7890A, Agilent technologies, Santa Clara, CA, USA) equipped with a mass spectrometer (5975C, Agilent Technologies, Santa Clara, CA, USA) and fitted with a DB-5 capillary column (30 m × 0.25 mm ID, 0.25 μm film thickness, J and W Scientific, Folsom, CA, U.S.). Helium at 1 mL·min^−^^1^ served as the carrier gas, and the GC inlet was set in splitless mode. The oven temperature was programmed to 40 °C for 2 min and then ramped to 220 °C at a rate of 6 °C·min^−1^ and held at 220 °C for 5 min. The mass spectrometer was operated in the electron impact ionization mode at a voltage of 70 eV. Mass spectra were taken over an m/z range of 30–400. Retention indices were calculated after analyzing C_8_-C_20_
*n*-alkane series under the same chromatographic conditions.

### 3.4. Identification of Volatile Compounds

These volatiles were further confirmed by matching their linear retention indices (LRIs) and odor descriptions in the literature [26,34,44]. The LRIs were computed according to the following Equation [47]:
(1)$$\mathrm{LRI}=100\times \left(\frac{\mathrm{Rt}\;\left(i\right)-\mathrm{Rt}\;\left(n\right)}{\mathrm{Rt}\;\left(n+1\right)-\mathrm{Rt}\;\left(n\right)+n}\right)$$
where Rt (i) is the retention time of the individual compound under investigation (i), Rt (n) and Rt (n + 1) refer to the retention times of *n*-alkanes (C_8_-C_22_; Supelco Analytical, Sigma, St. Louis, MO, USA) that elute before and after the target compound (i) for the same chromatographic conditions.

Aqueous 2, 4, 6-trimethylpyridine (40 mg·kg^−1^) was used as internal standard for quantitative analysis. The quantitative calculation (mg·kg^−1^) was based on:(2)$$\mathrm{Concentration}\;(\mathrm{each}\;\mathrm{compound})\;=\;\frac{\mathrm{Concentration}\;\left(\mathrm{internal}\;\mathrm{standard}\right)\times \mathrm{Peak}\;\mathrm{area}\;(\mathrm{each}\;\mathrm{compound})}{\mathrm{Peak}\;\mathrm{area}\;\left(\mathrm{internal}\;\mathrm{standard}\right)}$$

OAV for each volatile compound was calculated using the equation OAV = c/t, where c is the total concentration of the compound in the yogurt, and t is the odor threshold value. Compounds with OAV > 1 were considered as odor-active compounds [48].

### 3.5. Statistical Analyses

Analysis of variance (ANOVA) followed by Turkey and Dunnet’s multiple comparison test was adopted to evaluate the differences in volatile concentrations among yogurt samples. Heatmap and PCA were performed using MultiExp Simc-P 11.5 (Umetrics AB, Ume) to assess differences in volatile compounds and odor-active compounds, respectively. Unless specified, *p*-values of <0.05 were considered statistically significant.

## 4. Conclusions

In the present study, the influence of starter cultures on the volatile profiles of milks were investigated. A total of 53 volatile compounds and 31 odor-active compounds were detected in all samples. Ketones and hydrocarbons were the predominant volatile components in the milk before fermentation, while acids became the predominant volatiles in all fermented samples. (E,E)-2,4-decadienal, 2-nonanone,2-heptanone, and 2-undecanone with high OAVs were characteristic odor compounds in all samples. Fermentation increased OAVs of acids in samples. Compared with probiotics, there were greater abundance volatile compounds and odor-active compounds detected in fermented milks with STs2 and LB8. The heatmap and PCA showed a highly consistent distribution of volatile profiles based on the content and the OAVs. By means of heatmap and PCA, samples were divided into three kinds. STs2 and probiotics (L28, LB531, LC475, and LP45) belonged to one kind, and LB8 and BFM separately belonged to the other two kinds. The obtained data could help to better understand the influence of starter cultures on the odor quality of fermented milk and significant values for strain utilization and dairy development in dairy industry.

## Figures and Tables

**Figure 1 molecules-25-01129-f001:**
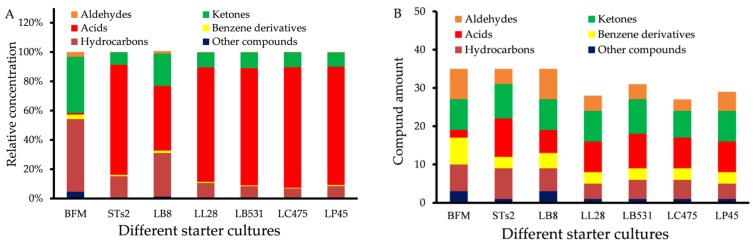
(**A**) Relative concentrations of different types of volatile compounds in milks fermented by different starter cultures. (**B**) Compound amount of different type volatile compounds in milks fermented by different starter cultures.

**Figure 2 molecules-25-01129-f002:**
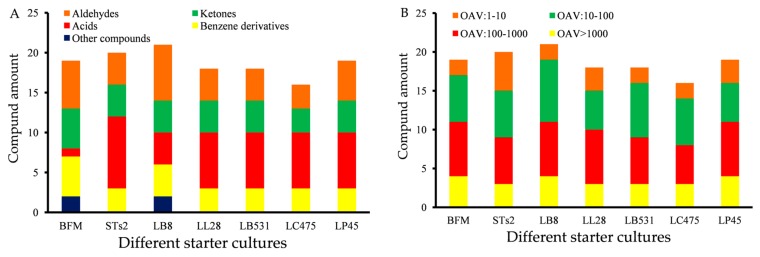
(**A**) Distribution of odor-active compounds in milks fermented by different starter cultures based on type of compounds. (**B**) Distribution of odor-active compounds in milks fermented by different starter cultures based on the level of odor activity values (OAVs).

**Figure 3 molecules-25-01129-f003:**
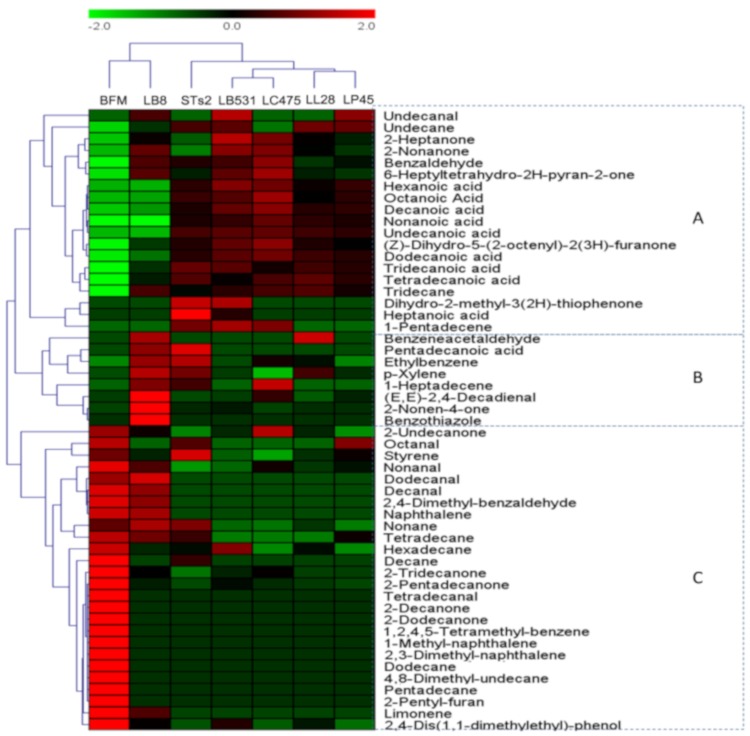
Heatmap of significantly different volatile compounds in milks fermented by different starter cultures based on the content. Green colors indicate that volatile compound levels were less than the mean levels, while red colors indicate that volatile compound levels were higher than the mean levels. Based on the content, volatile compounds were divided into three groups: A, B, and C.

**Figure 4 molecules-25-01129-f004:**
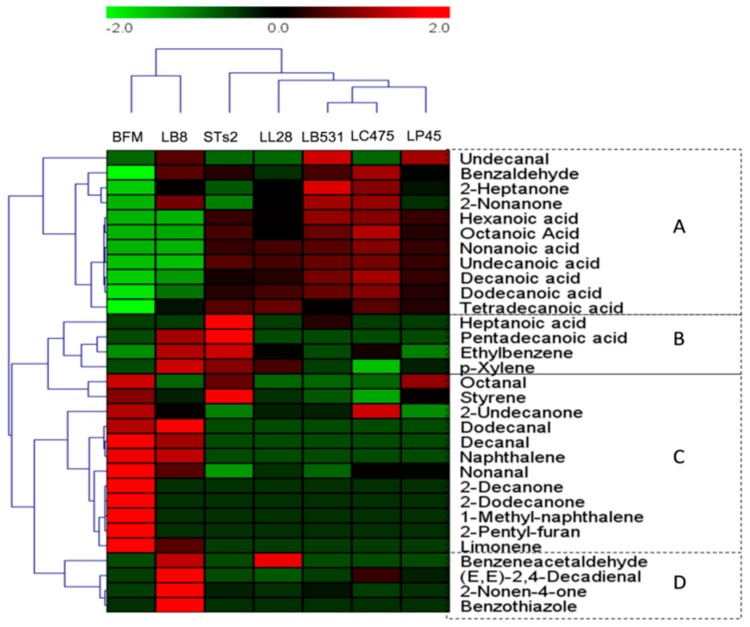
Heatmap of significantly different volatile compounds in the milks fermented by different starter cultures based on the OAVs. Green colors indicate that volatile compound levels were less than the mean levels, while red colors indicate that volatile compound levels were higher than the mean levels. Based on the OAVs, odor-active compounds were divided into four groups: A, B, C and D.

**Figure 5 molecules-25-01129-f005:**
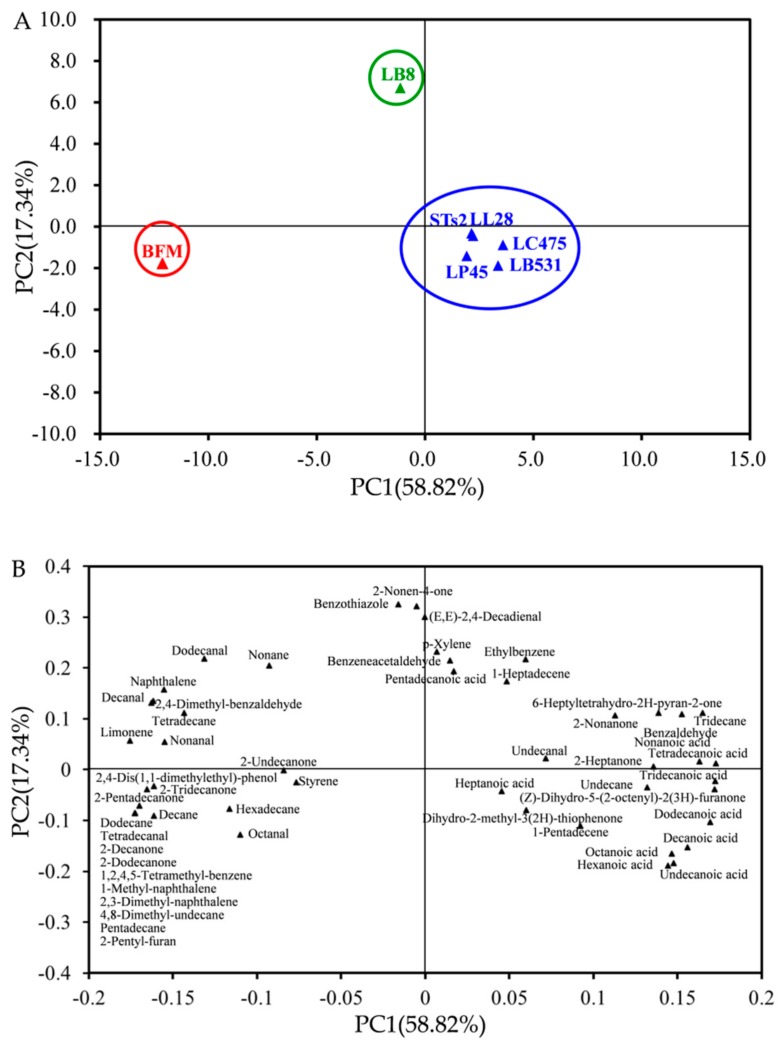
Principle component analysis (PCA) plots from (**A**) separation of milks fermented by different starter cultures based on the content. (**B**) Distribution of volatile compounds in milks fermented by different starter cultures based on content.

**Figure 6 molecules-25-01129-f006:**
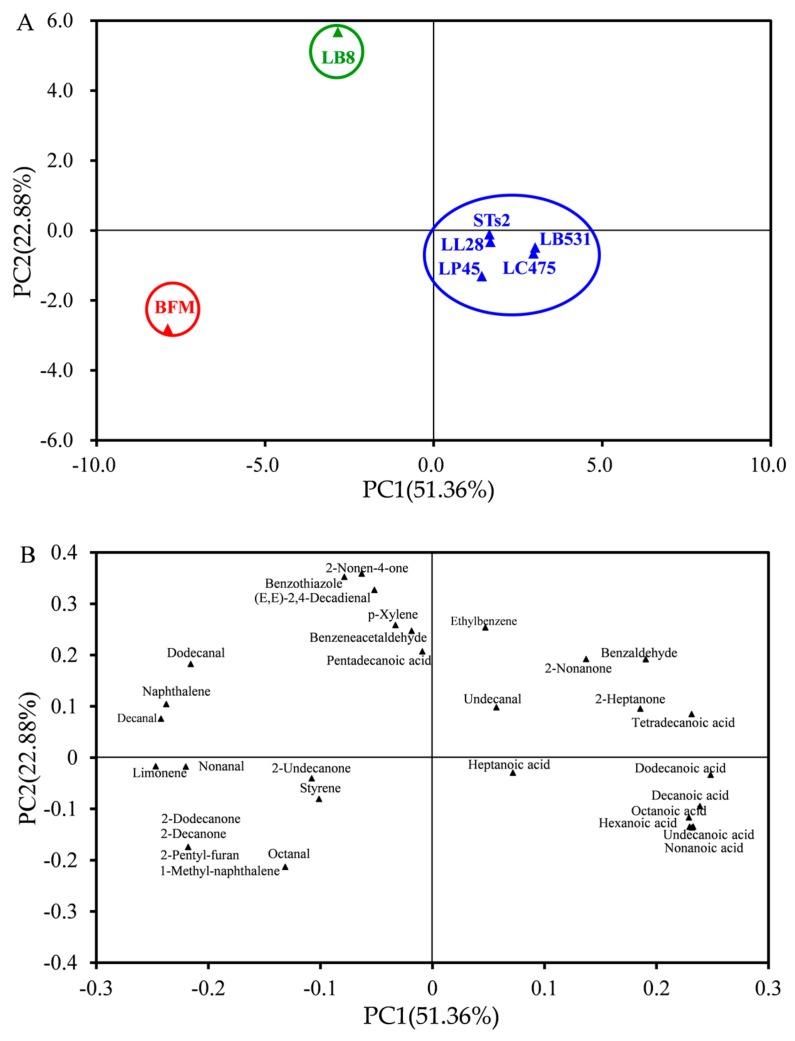
PCA plots from (**A**) separation of milks fermented by different starter cultures based on OAVs. (**B**) Distribution of volatile compounds in milks fermented by different starter cultures based on OAVs.

**Table 1 molecules-25-01129-t001:** Content of volatile compounds in the milks fermented with different starter cultures (n = 3).

RI	Compound Name	Content (mg·kg^−1^)						
		BFM	Traditional Starter Cultures		Probiotics			
			STs2	LB8	LL28	LB531	LC475	LP45
	Aldehydes							
955	Benzaldehyde	1.34 ± 0.18 ^a^	3.39 ± 0.41 ^c,d^	3.70 ± 0.26 ^d,e^	2.77 ± 0.13 ^b^	3.61 ± 0.42 ^b,c,d^	4.15 ± 0.40 ^e^	3.03 ± 0.37 ^b,c^
1004	Octanal	1.45 ± 0.13 ^c^	0.93 ± 0.10 ^a^	nd	nd	nd	nd	1.17 ± 0.14 ^b^
1045	Benzeneacetaldehyde	nd	nd	1.53 ± 0.08 ^a^	1.77 ± 0.23 ^b^	nd	nd	nd
1106	Nonanal	8.54 ± 2.01 ^e^	3.03 ± 0.44 ^a^	6.32 ± 0.73 ^d^	4.43 ± 0.44 ^a,b,c^	3.73 ± 0.26 ^a,b^	5.46 ± 0.48 ^c,d^	4.94 ± 0.63 ^b,c,d^
1206	Decanal	2.35 ± 0.07 ^b^	nd	1.74 ± 0.45 ^a^	nd	nd	nd	nd
1216	2,4-Dimethyl-benzaldehyde	6.62 ± 0.34 ^b^	nd	4.79 ± 0.49 ^a^	nd	nd	nd	nd
1308	Undecanal	nd	nd	2.15 ± 0.13 ^a^	nd	3.51 ± 0.32 ^c^	nd	2.98 ± 0.05 ^b^
1318	(E, E)-2,4-Decadienal	2.22 ± 0.28 ^a^	2.14 ± 0.47 ^a^	4.38 ± 0.35 ^c^	2.03 ± 0.23 ^a^	2.22 ± 0.21 ^a^	2.96 ± 0.43 ^b^	2.39 ± 0.45 ^a,b^
1410	Dodecanal	8.75 ± 1.14 ^a^	nd	10.99 ± 1.11 ^b^	nd	nd	nd	nd
1614	Tetradecanal	5.38 ± 0.98 ^a^	nd	nd	nd	nd	nd	nd
	Subtotal	36.64 ± 4.25 ^c^	9.49 ± 1.05 ^a^	35.60 ± 2.55 ^c^	11.00 ± 0.93 ^a,b^	13.07 ± 0.98 ^a,b^	12.57 ± 1.30 ^a,b^	14.52 ± 1.37 ^b^
	Ketones							
891	2-Heptanone	54.12 ± 2.90 ^a^	71.08 ± 6.41 ^b^	86.89 ± 7.95 ^c^	86.22 ± 3.86 ^c^	114.21 ± 8.57 ^d^	103.97 ± 12.81 ^d^	80.29 ± 5.13 ^b,c^
985	Dihydro-2-methyl-3(2H)-thiophenone	nd	1.58 ± 0.06 ^b^	nd	nd	1.43 ± 0.06 ^a^	nd	nd
1094	2-Nonanone	64.67 ± 4.98 ^a^	73.13 ± 5.67 ^a,b^	105.42 ± 9.98 ^d^	90.78 ± 4.23 ^c^	110.68 ± 9.65 ^d^	109.40 ± 12.54 ^d^	83.74 ± 6.34 ^b,c^
1130	2-Nonen-4-one	nd	2.23 ± 0.15 ^b^	29.74 ± 2.81 ^c^	1.72 ± 0.07 ^a^	3.55 ± 0.20 ^b^	nd	1.73 ± 0.06 ^a,b^
1199	2-Decanone	4.52 ± 0.40 ^a^	nd	nd	nd	nd	nd	nd
1296	2-Undecanone	88.21 ± 10.22 ^d^	58.96 ± 5.03 ^a^	73.87 ± 7.46 ^c^	68.83 ± 6.48 ^a,b^	68.47 ± 6.59 ^a,b^	91.10 ± 6.24 ^d^	57.76 ± 3.97 ^a^
1399	2-Dodecanone	1.98 ± 0.36 ^a^	nd	nd	nd	nd	nd	nd
1498	2-Tridecanone	112.01 ± 14.09 ^d^	46.51 ± 5.77 ^a^	66.08 ± 7.34 ^c^	55.40 ± 3.85 ^a,b^	60.19 ± 6.33 ^a,b^	67.07 ± 5.04 ^c^	55.27 ± 3.06 ^a,b^
1658	(Z)-Dihydro-5-(2-octenyl)-2(3H)- furanone	nd	10.24 ± 1.10 ^b^	6.90 ± 0.33 ^a^	10.08 ± 0.32 ^b^	12.45 ± 1.32 ^c^	13.96 ± 1.54 ^c^	8.89 ± 0.65 ^b^
1699	2-Pentadecanone	79.83 ± 8.94 ^d^	15.93 ± 1.85 ^a^	24.00 ± 2.06 ^b,c^	19.01 ± 1.54 ^a,b^	27.26 ± 2.83 ^b^	21.64 ± 3.02 ^a,b,c^	16.62 ± 1.61 ^a,b^
1710	6-Heptyltetrahydro-2H-pyran-2-one	34.58 ± 1.90 ^a^	61.24 ± 7.08 ^b^	77.49 ± 7.75 ^c^	61.32 ± 7.63 ^b^	78.10 ± 7.31 ^c^	87.35 ± 7.05 ^c^	58.03 ± 4.13 ^b^
	Subtotal	439.92 ± 33.78 ^b,c^	340.91 ± 22.28 ^a^	470.38 ± 38.65 ^c^	393.35 ± 15.37 ^a,b^	476.35 ± 37.46 ^c^	494.48 ± 44.77 ^c^	362.32 ± 20.43 ^c,a^
	Acids							
1008	Hexanoic acid	nd	104.57 ± 7.77 ^a,b^	nd	89.38 ± 8.03 ^a^	145.06 ± 12.16 ^c^	136.09 ± 8.64 ^c^	108.25 ± 15.84 ^b^
1086	Heptanoic acid	nd	3.84 ± 0.33 ^b^	nd	nd	1.15 ± 0.08 ^a^	nd	nd
1194	Octanoic acid	nd	444.97 ± 32.66 ^c,d^	37.17 ± 0.71 ^a^	341.76 ± 29.65 ^b^	494.19 ± 48.74 ^d^	630.76 ± 58.67 ^e^	420.94 ± 31.33 ^c^
1285	Nonanoic acid	nd	21.45 ± 2.02 ^a^	nd	22.83 ± 2.31 ^a^	23.24 ± 3.95 ^a^	26.95 ± 1.70 ^b^	21.54 ± 1.34 ^a^
1377	Decanoic acid	nd	1214.80 ± 130.83 ^b^	262.14 ± 28.60 ^a^	1277.22 ± 121.82 ^b^	1600.14 ± 117.62 ^c^	1803.23 ± 196.96 ^c^	1325.80 ± 182.11 ^b^
1473	Undecanoic acid	nd	26.64 ± 3.26 ^a,b^	nd	25.89 ± 2.58 ^a,b^	28.58 ± 0.79 ^c^	29.37 ± 2.52 ^c^	24.02 ± 2.86 ^a^
1568	Dodecanoic acid	10.92 ± 2.43 ^a^	780.44 ± 42.27 ^c^	349.29 ± 36.33 ^b^	842.41 ± 61.57 ^c,d^	926.22 ± 83.22 ^d,e^	990.58 ± 67.42 ^e^	782.37 ± 72.81 ^c^
1666	Tridecanoic acid	nd	7.47 ± 0.24 ^d^	4.42 ± 0.37 ^a^	6.72 ± 0.69 ^b,c,d^	7.21 ± 0.64 ^c,d^	5.79 ± 0.79 ^b^	6.20 ± 0.81 ^b,c^
1765	Tetradecanoic acid	0.98 ± 0.00 ^a^	399.41 ± 35.95 ^d^	275.51 ± 28.99 ^b^	410.76 ± 45.74 ^d^	316.44 ± 25.11 ^b,c^	390.32 ± 33.27 ^d^	354.34 ± 28.66 ^c,d^
1863	Pentadecanoic acid	nd	13.06 ± 2.45 ^b^	9.42 ± 0.32 ^a^	nd	nd	nd	nd
	Subtotal	11.90 ± 2.43 ^a^	3016.65 ± 246.12 ^c^	937.95 ± 77.39 ^b^	3016.97 ± 252.14 ^c^	3542.22 ± 184.24 ^d^	4013.10 ± 363.87 ^e^	3043.45 ± 264.52 ^c^
	Hydrocarbons							
900	Nonane	539.67 ± 72.04 ^b^	576.16 ± 35.13 ^b,c^	621.57 ± 57.82 ^c^	376.36 ± 38.32 ^a^	322.18 ± 16.95 ^a^	306.53 ± 17.14 ^a^	297.29 ± 30.21 ^a^
1000	Decane	1.11 ± 0.15 ^b^	0.39 ± 0.05 ^a^	nd	nd	nd	nd	nd
1100	Undecane	nd	1.54 ± 0.18 ^c^	0.89 ± 0.13 ^b^	1.76 ± 0.15 ^c^	1.66 ± 0.10 ^c^	0.53 ± 0.05 ^a^	1.72 ± 0.42 ^c^
1200	Dodecane	1.35 ± 0.10 ^a^	nd	nd	nd	nd	nd	nd
1224	4,8-Dimethyl-undecane	1.09 ± 0.16 ^a^	nd	nd	nd	nd	nd	nd
1300	Tridecane	nd	1.37 ± 0.05 ^a^	1.78 ± 0.17 ^b^	1.83 ± 0.18 ^b^	1.62 ± 0.27 ^a,b^	1.76 ± 0.18 ^b^	1.52 ± 0.16 ^a,b^
1400	Tetradecane	3.45 ± 0.26 ^c^	2.01 ± 0.48 ^a,b^	2.62 ± 0.31 ^b,c^	nd	nd	nd	1.52 ± 1.11 ^a^
1494	1-Pentadecene	nd	5.05 ± 0.93^a^	nd	nd	6.16 ± 0.58 ^b^	5.28 ± 0.39 ^a^	nd
1500	Pentadecane	8.24 ± 0.61 ^a^	nd	nd	nd	nd	nd	nd
1600	Hexadecane	12.68 ± 0.87 ^c^	4.67 ± 1.48 ^a^	3.70 ± 0.27 ^a^	4.72 ± 0.69 ^a^	10.10 ± 1.14 ^b^	nd	nd
1692	1-Heptadecene	nd	0.67 ± 0.12 ^a^	0.93 ± 0.15 ^b^	nd	nd	1.20 ± 0.21 ^c^	nd
	Subtotal	567.59 ± 72.94 ^c^	591.86 ± 36.86 ^c^	631.48 ± 58.40 ^c^	384.67 ± 38.68 ^b^	341.73 ± 16.83 ^a,b^	315.30 ± 17.85 ^a,b^	302.05 ± 30.32 ^a^
	Benzene derivatives							
852	Ethylbenzene	3.98 ± 0.41 ^a^	6.09 ± 0.44 ^c^	5.90 ± 0.35 ^c^	4.80 ± 0.51 ^b^	4.39 ± 0.20 ^ab^	5.04 ± 0.17 ^b^	4.01 ± 0.64 ^a^
860	p-Xylene	13.19 ± 1.68 ^b^	17.79 ± 1.37 ^c^	19.35 ± 1.67 ^c^	16.67 ± 1.13 ^c^	13.58 ± 1.35 ^b^	10.50 ± 1.48 ^a^	14.00 ± 1.65 ^b^
884	Styrene	12.71 ± 1.97 ^d^	15.10 ± 1.15 ^e^	9.41 ± 0.84 ^b,c^	9.13 ± 1.09 ^b,c^	8.04 ± 1.09 ^a,b^	6.49 ± 0.41 ^a^	10.49 ± 1.57 ^c^
1118	1,2,4,5-Tetramethyl-benzene	0.99 ± 0.15 ^a^	nd	nd	nd	nd	nd	nd
1183	Naphthalene	1.14 ± 0.07 ^b^	nd	0.97 ± 0.15 ^a^	nd	nd	nd	nd
1289	1-Methyl-naphthalene	3.49 ± 0.15 ^a^	nd	nd	nd	nd	nd	nd
1417	2,3-Dimethyl-naphthalene	1.05 ± 0.04 ^a^	nd	nd	nd	nd	nd	nd
	Subtotal	36.56 ± 3.83 ^c^	38.98 ± 2.09 ^c^	35.63 ± 2.21 ^c^	30.59 ± 1.98 ^b^	26.02 ± 0.47 ^a^	22.03 ± 1.34 ^a^	28.49 ± 1.39 ^a,b^
	Other compounds							
991	2-Pentyl-furan	2.19 ± 0.38 ^a^	nd	nd	nd	nd	nd	nd
1029	Limonene	3.67 ± 0.69 ^b^	nd	1.64 ± 0.17 ^a^	nd	nd	nd	nd
1218	Benzothiazole	nd	nd	0.63 ± 0.07 ^a^	nd	nd	nd	nd
1515	2,4-Dis (1,1-dimethylethyl)-phenol	46.85 ± 5.33 ^d^	18.45 ± 0.66 ^a,b^	25.74 ± 3.47 ^c^	23.37 ± 3.67 ^b,c^	28.34 ± 2.59 ^c^	17.82 ± 0.6 ^a^	16.25 ± 1.27 ^a^
	Subtotal	52.71 ± 6.06 ^d^	18.45 ± 0.66 ^ab^	28.00 ± 3.67 ^c^	23.37 ± 3.67 ^b,c^	28.34 ± 2.59 ^c^	17.82 ± 0.62 ^a,b^	16.25 ± 1.27 ^a^
	Total	1145.32 ± 116.21 ^a^	4016.34 ± 300.98 ^d^	2125.68 ± 166.459 ^b^	3859.96 ± 228.49 ^c^	4427.71 ± 229.48 ^d^	4875.29 ± 427.64 ^d^	3767.08 ± 272.01 ^c^

RI: retention index; Nd: no detection; mean values in the same row with different letters indicate the significant differences between clusters (*p* < 0.05).

**Table 2 molecules-25-01129-t002:** Odor activity values (OAVs) of volatile compounds in the milks fermented with different starter cultures.

Compound Name	Odor Description	Threshold (mg·kg^−1^)	OAVs						
			BFM	STs2	LB8	LL28	LB531	LC475	LP45
Aldehydes									
Benzaldehyde	Almond, burnt sugar [26]	0.35	3.8	9.7	10.6	7.9	10.3	11.8	8.7
Octanal	Fat, soap, lemon, green [26]	0.0009	1608.2	1035.7	0	0	0	0	1302.1
Benzeneacetaldehyde	Sweet, flora [44]	0.004	0	0	383.3	442.7	0	0	0
Nonanal	Sweet, floral, citrus, grass [34]	0.04	213.4	75.8	157.9	110.8	93.3	136.6	123.6
Decanal	Fatty [26]	0.03	78.3	0	57.9	0	0	0	0
Undecanal	Fatty [26]	0.01	0	0	215.5	0	351.2	0	298.3
(E, E)-2,4-Decadienal	Fatty, green [44]	0.0001	22156.6	21353.0	43761.4	20301.9	22190.4	29582.3	23910.3
Dodecanal	Soapy, waxy, citrus [34]	0.0015	5836.5	0	7324.5	0	0	0	0
Ketones									
2-Heptanone	Fresh cream [34]	0.14	386.6	507.7	620.6	615.8	815.8	742.6	573.5
2-Nonanone	Grassy, fruity, floral [44]	0.08	808.4	914.1	1317.7	1134.8	1383.6	1367.5	1046.8
2-Nonen-4-one	Fruity, floral [26]	0.0009	0.0	2476.8	33047.3	1915.2	3939.0	0	1917.5
2-Decanone	Sweet, waxy [26]	0.08	56.5	0	0	0	0	0	0
2-Undecanone	Orange, grass, fresh [34]	0.08	1102.6	737.0	923.3	860.3	855.8	1138.8	722.0
2-Dodecanone	Orange, grass, fresh [44]	0.08	24.8	0	0	0	0	0	0
Acids									
Hexanoic acid	Spicy, rancid, floral [34]	1.84	0	56.8	0	48.6	78.8	74.0	58.8
Heptanoic acid	Rancid [44]	3	0	1.3	0	0	0.4	0	0
Octanoic acid	Chess, sweaty [34]	1.9	0	234.2	19.6	179.9	260.1	332.0	221.5
Nonanoic acid	Rancid [44]	3	0	7.2	0	7.6	7.7	9.0	7.2
Decanoic acid	Rancid [44]	3	0	404.9	87.4	425.7	533.4	601.1	441.9
Undecanoic acid	Rancid [44]	10	0	2.7	0	2.6	2.9	2.9	2.4
Dodecanoic acid	Rancid [44]	10	1.1	78.0	34.9	84.2	92.6	99.1	78.2
Tridecanoic acid	Rancid [44]	10	0	0.7	0.4	0.7	0.7	0.6	0.6
Tetradecanoic acid	Rancid [44]	10	0.1	39.9	27.6	41.1	31.6	39.0	35.4
Pentadecanoic acid	Rancid [44]	10	0	1.3	0.9	0	0	0	0
Benzene derivatives									
Ethylbenzene	Gasoline [44]	0.2	19.9	30.4	29.5	24.0	22.0	25.2	20.0
p-Xylene	Phenolic [44]	1	13.2	17.8	19.4	16.7	13.6	10.5	14.0
Styrene	Sweet, balsamic [44]	0.05	254.2	302.0	188.1	182.6	160.9	129.8	209.8
Naphthalene	Phenolic [44]	0.006	190.6	0	161.8	0	0	0	0
1-Methyl-naphthalene	Phenolic [44]	0.0075	465.1	0	0	0	0	0	0
Other compounds									
2-Pentyl-furan	Green, fat [34]	0.006	364.9	0	0	0	0	0	0
Limonene	Orange, mint [26]	0.2	18.4	0	8.2	0	0	0	0
Benzothiazole	Meaty, vegetative [34]	0.08	0	0	7.9	0	0	0	0

Odor description and threshold were provided in references [26,34,44]. BFM: milk before fermentation.

**Table 3 molecules-25-01129-t003:** Fermentation parameters of different starter cultures.

Starter Cultures and Model	Abbreviation	Fermentation Temperature (°C)	Concentration of Inoculums (cfu·kg^−1^)
*Streptococcus thermophilus* s2	STs2	42	3 × 10^8^
*Lactobacillus bulgaricus* 8	LB8	42	6 × 10^8^
*Lactobacillus lactis* 28	LL28	37	3 × 10^8^
*Lactobacillus bifidus* 45	LB45	37	3 × 10^8^
*Lactobacillus casei* 475	LC475	37	3 × 10^8^
*Lactobacillus plantarum* 531	LP531	37	6 × 10^8^

Note: the starter cultures and the fermentation parameter of different starter culture were provided by Hebei Inatural Bio-Tech Co., Ltd. (Hebei, China).

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
