# Peer review of "Evaluation of Volatile Compounds in Milks Fermented Using Traditional Starter Cultures and Probiotics Based on Odor Activity Value and Chemometric Techniques"

_molecules, 2020, doi:10.3390/molecules25051129_

Round 1

Reviewer 1 Report

This paper studies the interspecies variability in lactic acid bacteria in relation to the production of volatile compounds during yogurt fermentation. The structure and the contents are appropriate and the only detail to note regarding the format is that the authors should improve the English of the document. The equipments and analytical techniques used, as well as the chemometris applied are of great interest and perfectly suitable for addressing the objectives of the study (I really found very useful the heatmap). I will recommend the acceptance of the manuscript for publication by Molecules, after correction and response to the following issues.

Lines 13 and 77: Correct the name: “Streptococcus thermophiles”

Line 279: based “on” the content

Line 405: Why was the milk fortified with 6% sugar?

Table 3: How can the results be compared if the population of microorganisms is different in each of the inocula evaluated? Was the volume of inoculation adjusted to this g/kg? Please, if so, show this data or include a sentence to reference it. Why have you decided to use these units and not report the cfu/kg?

The results of the study have a high potential for further research. Have the authors considered the possibility of using mixed cultures for the production of yoghurts?

In the conclusion, the authors should highlight the direct impact and most relevant use of this study for the industry.

Author Response

Lines 13 and 77: Correct the name: “Streptococcus thermophiles”

Line 13,77 and 242: we corrected the name “Streptococcus thermophiles” to “Streptococcus thermophilus”

Line 279: based “on” the content

Line 279: We added “on” after “based” in the article.

Line 405: Why was the milk fortified with 6% sugar?

Line 405: Provide a carbon source for different starter cultures fermentation to ensure the effectiveness of fermentation.

Table 3: How can the results be compared if the population of microorganisms is different in each of the inocula evaluated? Was the volume of inoculation adjusted to this g/kg? Please, if so, show this data or include a sentence to reference it. Why have you decided to use these units and not report the cfu/kg?

We corrected the unit of concentration of inoculums, and changed it from “g/kg” to “cfu/kg”

We showed the detail concentration of inoculums with cfu/kg in Table 3.

The results of the study have a high potential for further research. Have the authors considered the possibility of using mixed cultures for the production of yoghurts?

We are considering further research on the use of mixed strains to improve the quality of yogurt fermentation. If possible, we are very hope to industrialize the production of mixed strains in the yogurt industry.

In the conclusion, the authors should highlight the direct impact and most relevant use of this study for the industry.

In the conclusion, we added “The obtained data could help to better understand the influence of starter cultures on the odor quality of fermented milk, and significant value for strain utilization and dairy development in dairy industry.” to highlight the direct impact and most relevant use of this study for the industry.

Lines 13 and 77: Correct the name: “Streptococcus thermophiles”

Line 13,77 and 242: we corrected the name “Streptococcus thermophiles” to “Streptococcus thermophilus”

Line 279: based “on” the content

Line 279: We added “on” after “based” in the article.

Line 405: Why was the milk fortified with 6% sugar?

Line 405: Provide a carbon source for different starter cultures fermentation to ensure the effectiveness of fermentation.

Table 3: How can the results be compared if the population of microorganisms is different in each of the inocula evaluated? Was the volume of inoculation adjusted to this g/kg? Please, if so, show this data or include a sentence to reference it. Why have you decided to use these units and not report the cfu/kg?

We corrected the unit of concentration of inoculums, and changed it from “g/kg” to “cfu/kg”

We showed the detail concentration of inoculums with cfu/kg in Table 3.

The results of the study have a high potential for further research. Have the authors considered the possibility of using mixed cultures for the production of yoghurts?

We are considering further research on the use of mixed strains to improve the quality of yogurt fermentation. If possible, we are very hope to industrialize the production of mixed strains in the yogurt industry.

In the conclusion, the authors should highlight the direct impact and most relevant use of this study for the industry.

In the conclusion, we added “The obtained data could help to better understand the influence of starter cultures on the odor quality of fermented milk, and significant value for strain utilization and dairy development in dairy industry.” to highlight the direct impact and most relevant use of this study for the industry.

Reviewer 2 Report

The authors presented a large characterisation of the volatiles produced by lactic acid bacteria in yougurt fermentation that deserves to be published. However, some issues should be adressed:

Main issues

Authors separate starins according to probiotic activity but this was not evaluated. Even S. thermophilus is considered probiotic by some reports. So, it is more correct to remove the separation among strains according to probiotic activity. Just mention that milk was fermented by several lactic acid bacteria. The OAV is an approximate measure of the expected odour. Authors did not perform sensory analysis, so it is not correct to infer sensory descriptors from OAV data. For instance, in conclusions authors should state that results are preliminar concerning sensory description, probably the yogurts smell all to the same despite the different aroma composition. Only LB8 was separated from the others in PCA, so S. thermophilus behaved like the other probiotic strains. So probiotic activity has nothing to do with flavour production and this should be mentioned if authors feel that probiotic activity should be retained.

Minor corrections

Replace thermophile by thermophilus throughout the text. Figures: replace amount by concentration, where necessary. Line 96: what is the meaning of "high n amounts"? Place figures after reference in the text (figs. 2, 3 and 6).

Author Response

Main issues

Authors separate starins according to probiotic activity but this was not evaluated. Even S. thermophilus is considered probiotic by some reports. So, it is more correct to remove the separation among strains according to probiotic activity. Just mention that milk was fermented by several lactic acid bacteria. The OAV is an approximate measure of the expected odour. Authors did not perform sensory analysis, so it is not correct to infer sensory descriptors from OAV data. For instance, in conclusions authors should state that results are preliminar concerning sensory description, probably the yogurts smell all to the same despite the different aroma composition. Only LB8 was separated from the others in PCA, so S. thermophilus behaved like the other probiotic strains. So probiotic activity has nothing to do with flavour production and this should be mentioned if authors feel that probiotic activity should be retained.

We corrected the unit of concentration of inoculums, and changed it from “g/kg” to “cfu/kg” in table 3. We also added discussion about probiotics activity in manuscript. The added content is “S. thermophilus is multifunctional lactic acid bacteria, also used as probiotics in some studies. As shown in heatmap and PCA, S. thermophilus behaved more like the other probiotic strains. The different distribution of odor components may be caused by the probiotics activity. The inherent relationship between probiotic activity and volatile components needs further study.”

We corrected the inappropriate views about OAVs in the conclusions.

Minor corrections

Replace thermophile by thermophilus throughout the text.

Replaced thermophile by thermophilus Line 13 77 and 242

Figures: replace amount by concentration, where necessary.

Replaced “amount” by “concentration” in fig.1-A.

Line 96: what is the meaning of "high n amounts"?

Changed "high n amounts" into "high amounts"

Place figures after reference in the text (figs. 2, 3 and 6). 

We put the figures in the text according to the manuscript format of journal.

Reviewer 3 Report

Please find my comments in the attached file

Author Response

General Comments

1) Yoghurt is the product that results from the synergistic action of at least the two symbiotic cultures of Streptococcus thermophilus and Lactobacillus delbrueckii subsp. Bulgaricus on milk. Any other milk product made with one or more different cultures is simply called fermented milk. Therefore, to be more precise the title of your research work should change from ‘Evaluation of Volatile Compounds in the Yogurts Fermented Using Traditional …’ to ‘Evaluation of Volatile Compounds in Milks Fermented Using Traditional…’. Please, change also in the text.

We changed the title from “Evaluation of Volatile Compounds in the Yogurts Fermented Using Traditional Starter Cultures and Probiotics Based on Odor Activity Value and Chemometric Techniques” to “Evaluation of Volatile Compounds in Milks Fermented Using Traditional Starter Cultures and Probiotics Based on Odor Activity Value and Chemometric Techniques ”

In addition, we also changed “yogurt” to”milk” in the whole text.

2) I am impressed that too many volatile components resulted from fermented milk products aged just one day [Line 409]. You did not study the evolution of these compounds throughout the shelf life of the products. However, you discuss about proteolysis and lipolysis citing other papersconcerning cheeses! [Lines 297-298]. The truth is that the composition of volatile compounds inyoghurt depends on the species of microorganism used either alone or in combination but, the contribution of lipolysis to the flavor of yoghurt is limited, compared with the long-ripening cheeses.

We checked the reference again which come from cheeses. We changed the reference related to yogurt [40], and corrected the cited content.

New content is ”The hydrolysis of triacylglycerols under the action of lactic acid bacteria is the main pathway for the formation of carboxylic acids [40].”

3) In general, you cite articles with research on cheese or other product, e.g. references 38, 39, 40, 43, 25. Please refer mainly to articles concerning volatiles in yoghurt and rewrite the discussion.

We changed references 38, 39, 40, 43, and 25 to new references which closely related to yogurt and its fermented milk. The new reference No is 40 (old 38), 41 (old 39, 40), 39(old 43) and 27 (old 25). We also rewrote the discussion about these reference.

4) The conclusion in the abstract [Lines 26-28], is not safe. In my opinion, you cannot differentiate the fermented products by the starters used since there are so many different subspecies used and the volatile compounds change throughout storage. In contrast, conclusion in lines 467-468 is OK.

In abstract, we changed “The results demonstrated that the content and OAVs of volatile compounds enabled a good differentiation of yogurts fermented by different starter cultures.” into “The results could help to better understand the influence of starter cultures on the odor quality of milks.”

Specific comments 

Lines 56-57: …there are few studies… References should be added. 

We added to new references 24 and 25 in the article.

Line 144: … the presence of mycelium…. In yoghurt?  

This sentence come from reference cited from cheese article. So we changed to new reference about yogurt. “In addition, changes in ketones during fermentation is affected by textural properties, stabilizing agents, and the concentration of fatty acids [39]”.

Lines 304-306: What is the source of hydrocarbons found in raw milk?

We added “Hydrocarbons are generally the secondary products of lipid autoxidation in raw milk [41]”

Round 2

Reviewer 2 Report

The article may be accepted in the current form.

Reviewer 3 Report

All my comments/corrections were taken into consideration by the authors and the manuscript was revised accordingly.